# Dynamic organization of Herpesvirus glycoproteins on the viral envelope revealed by super-resolution microscopy

Frauke Beilstein[1], Gary H. Cohen[2], Roselyn J. Eisenberg[3], Valérie Nicolas[4], Audrey Esclatine[1], David Pasdeloup[1,5]*

1 Institute for Integrative Biology of the Cell (I2BC), CEA, CNRS, Univ. Paris-Sud, Université Paris-Saclay, Gif-sur-Yvette cedex, France, 2 Department of Microbiology, School of Dental Medicine, University of Pennsylvania, Philadelphia, Pennsylvania, United States of America, 3 Department of Pathobiology, School of Veterinary Medicine, University of Pennsylvania, Philadelphia, Pennsylvania, United States of America, 4 IPSIT, Microscopy facility, University of Paris-Sud, Châtenay-Malabry, France, 5 Laboratory of Biology of Avian Viruses, UMR1282 ISP, INRA Centre Val-de-Loire, Nouzilly, France

* david.pasdeloup@inra.fr

**Data Availability Statement:** All relevant data are within the manuscript and its Supporting Information files.

**Funding:** This work was funded by the Agence Nationale pour la Recherche (ANR) under grant

## Abstract

The processes of cell attachment and membrane fusion of Herpes Simplex Virus 1 involve many different envelope glycoproteins. Viral proteins gC and gD bind to cellular receptors. Upon binding, gD activates the gH/gL complex which in turn activates gB to trigger membrane fusion. Thus, these proteins must be located at the point of contact between cellular and viral envelopes to interact and allow fusion. Using super-resolution microscopy, we show that gB, gH/gL and most of gC are distributed evenly round purified virions. In contrast, gD localizes essentially as clusters which are distinct from gB and gH/gL. Upon cell binding, we observe that all glycoproteins, including gD, have a similar ring-like pattern, but the diameter of these rings was significantly smaller than those observed on cell-free viruses. We also observe that contrary to cell-free particles, gD mostly colocalizes with other glycoproteins on cell-bound particles. The differing patterns of localization of gD between cell-free and cell-bound viruses indicates that gD can be reorganized on the viral envelope following either a possible maturation of the viral particle or its adsorption to the cell. This redistribution of glycoproteins upon cell attachment could contribute to initiate the cascade of activations leading to membrane fusion.

## Author summary

The envelopes of Herpesvirus particles contain a variety of different proteins that allow them to infect specific cell types. An essential core set of these proteins is designed to allow viral entry into the cell after adsorption by binding to specific receptors and ultimately inducing fusion between the viral and cellular membranes in a regulated way through a succession of interactions between receptor-binding and fusion-triggering viral proteins. We have identified here for the first time the localization patterns of these essential proteins at the surface of purified virions and we describe how their localization

number ANR-14-CE14-0022. The funders had no role in study design, data collection and analysis, decision to publish, or preparation of the manuscript.

**Competing interests:** The authors have declared that no competing interests exist.

changes after cell attachment. These results illustrate how the dynamics of viral proteins at the surface of the viral particle could participate in optimizing the all-important process of cell binding and membrane fusion.

## Introduction

Herpesviruses encode a large number of different glycoproteins some of whose functions remain unclear. For Herpes Simplex Virus 1 (HSV-1), the core set of proteins required for attachment and entry are gD, gH, gL and gB. gD interacts with receptors such as HVEM (Herpesvirus Entry Mediator) and Nectin-1 and 2 [1–5], and activates the gH/gL complex and gB to trigger membrane fusion [6, 7]. Thus, these proteins must be spatially close on the virus envelope to allow this chain of activations. In particular, direct interaction of gB and gH/gL is required for fusion [8, 9]. In addition to these core proteins, glycoprotein gC engages heparan sulfates to facilitate viral particle attachment prior to the binding of gD to its specific receptors [10]. gC is non-essential in cell culture since binding to heparan sulfates can also be mediated by gB [11]. In the absence of precise data, it has generally been assumed that glycoproteins are randomly distributed over the viral envelope, although an uneven distribution at distinct poles of the particles has been described [12]. Observation of the actual organization of a selected glycoprotein on the viral envelope has been possible so far only by immuno-electron microscopy (immuno-EM) [13] and fluorescence microscopy. Immuno-EM allows the visualization of specific protein types on the particle but uses conditions of sample preparation which can alter antigenic properties and possibly change distributions of the protein. In addition, the relatively small number of particles which can be reliably analyzed by immuno-EM often limits the strength of the interpretation. This limit also applies to cryo-EM which cannot distinguish between the different types of glycoproteins although it allows observation of proteins in their native state at a relatively high resolution. Although fluorescence microscopy allows for the analysis of a great number of particles, it is limited by the diffraction limit of light to a resolution of around 200 nm, which is roughly the size of the HSV-1 virion. Super-resolution fluorescence microscopy (or nanoscopy) is a new and powerful tool which allows the visualization of the organization of a specific glycoprotein at the surface of viral particles. It combines the advantage of using specific antibodies to target a given protein on a large number of particles with an achievable resolution compatible with the identification of sub-structures within the viral particle [14–16]. Here we used stimulated emission-depletion (STED) microscopy to analyze the organization of gC, gD, gB and gH/gL at the surface of HSV-1 particles with a lateral resolution ranging from 44 to 60 nm. We describe and characterize the patterns of glycoprotein organization at the surface of both free and cell-bound virions. Our results raise the possibility that glycoproteins or a subset of glycoproteins are reorganized at the surface of the herpesvirus particle following a possible maturation process or as a direct consequence of cell binding.

## Results

### Super-resolution microscopy reveals a variety of glycoprotein distributions at the surface of purified viral particles

HSV-1 virions (unmodified 17+ strain) purified on a Ficoll gradient and attached to glass coverslips were visualized using confocal microscopy with or without a gated STED (gSTED) set-up (see Methods for details). Virions labeled with monoclonal antibody (mAb) LP11 directed against gH/gL and observed in diffraction-limited mode showed an overall picture consisting of fluorescent spots that were uniformly round, large in size (average of 454 +/- 50 nm, n = 67)

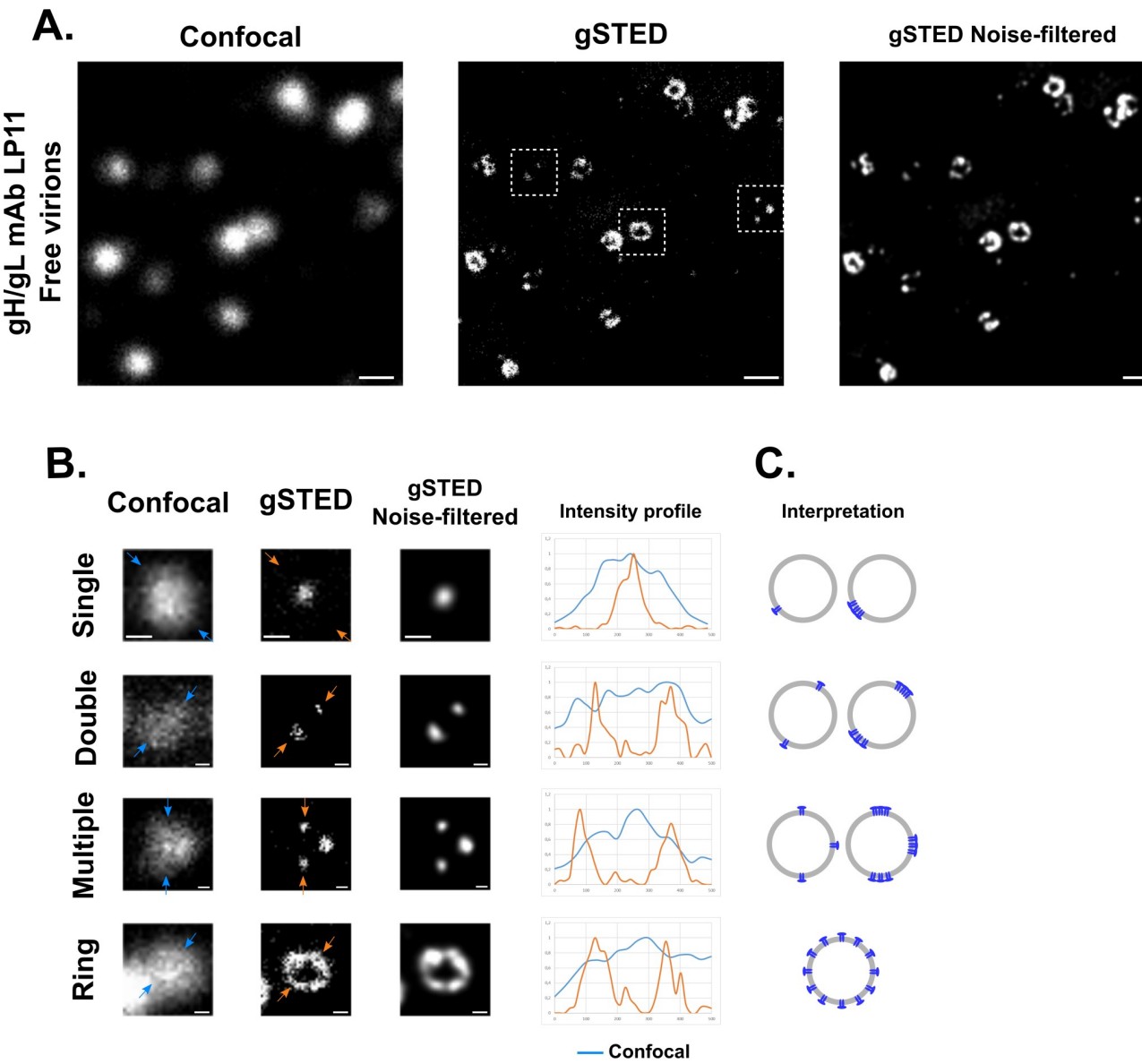

**Fig 1. gSTED reveals various features of glycoprotein organization at the surface of viral particles.** (A) Purified WT 17+ virions were attached onto a glass coverslip and labeled with mAb LP11 against gH/gL and an Oregon green 488-labeled secondary antibody. Images were acquired using the diffraction-limited confocal mode or with a gated STED set-up (gSTED). Images were further processed for noise-filtering using Huygens software algorithms. Scale bars: 500 nm. (B) Different patterns of glycoprotein organization were observed in the gSTED mode, which were not resolvable in the confocal mode. From top to bottom these were: single, double, and multiple spots, and rings. Three of the patterns correspond to particles boxed in (A). The single spot image is of a particle labeled with gC-specific mAb IC8 and is included here for illustrative purposes. Each different pattern is shown in the confocal and in the gSTED mode with corresponding noise-filtered images. Arrows delineate the line (500 nm long) used for profile analysis of normalized intensity. The blue line and the orange line in the graph correspond to the intensity profiles observed in the confocal and in the gSTED modes respectively. Analysis was carried out on raw, unprocessed images. Scale bars: 100 nm. (C) Schematic illustration as a 2D projection of the different patterns described in (B). The number of proteins per cluster (one or more) is unknown.

and varying in intensity ([Fig 1A](), Confocal). When the gSTED set-up was applied, the large uniform spots resolved to show more varied and better defined features ([Fig 1A](), gSTED). These features could be further refined by applying a noise-filtering algorithm ([Fig 1A](), gSTED Noise-filtered). The features observed consisted essentially of single, double or multiple spots,

or rings of glycoproteins (Fig 1B), features which were previously impossible to visualize by conventional methods of fluorescence microscopy. When "multiple spots" were observed, they were in overwhelming majority composed of three or four spots, with few occurrences as high as seven or eight spots. We interpret the spots as comprising a single glycoprotein or a group of glycoproteins localized together in a distinct location (Fig 1C). Rings are most likely multiple glycoproteins or groups of glycoproteins distributed all around the viral envelope and appearing as a ring in a confocal section (our STED set-up does not resolve in z).

As shown in Fig 1B, these distinct types of localization were indiscernible by conventional confocal microscopy. The resolution obtained with our gSTED set-up was estimated by FWHM (Full Width at Half-Maximum) analysis on immune complexes at 60 nm on average (n = 6) (S1 Fig). We thus conclude that the distribution of glycoproteins gH/gL may differ between individual virions, and we investigated whether similar patterns were also observed with other glycoproteins.

## Glycoprotein gD but not gB and gH/gL localizes differently at the surface of free and cell-bound virions

The patterns of distribution of glycoproteins gC, gD, gH/gL and gB were determined on WT virions which were either free or bound to cell surfaces. We used WT virions to ensure that glycoprotein distributions were not influenced by the presence of a tag attached to any viral protein. In order to test for the influence of temperature and cell type on glycoprotein localization, two sets of experiments were carried out. In the first, virions were attached to glass coverslips at room-temperature or incubated with human fibroblasts (HFFF2 cells) at 4°C to allow cell attachment while preventing membrane fusion. In the second set of experiments, virions were attached to glass coverslips at 4°C or bound to HeLa cells, also at 4°C. In all cases, virions were left to attach for 30 mins and were then washed, fixed and stained using one monoclonal (mAb) and one polyclonal (pAb) antibody for each glycoprotein. Observation of a large number of particles by gSTED revealed that the distribution of glycoproteins gB and gH/gL on free and cell-bound virions was essentially a ring-like distribution for ~60% or more of observed particles in both sets of experiments (Fig 2, red bars). This result was consistent between polyclonal and monoclonal antibodies and was similar for HFFF-bound (Fig 2A) or HeLa-bound virions (Fig 2B). gB was more commonly observed as rings on cell-bound viruses than on cell-free viruses, in particular with mAb SS63 on HeLa-bound particles where this observation was statistically significant. In contrast, receptor-binding glycoproteins gD and gC showed different patterns (see below for gC). Only ~40% or fewer of free virions displayed a ring-like distribution for gD regardless of the antibody or the incubation temperature used, indicating that the glycoprotein is preferentially organized as clusters on the viral envelope. However, in all cases there was a significant increase in the proportion of cell-bound particles that displayed ring-like localization for gD when compared to free virions: 73% of HFFF-bound particles as opposed to 40% for pAb R8 and 94% as opposed to 36% for mAb MC23, and 78% of HeLa-bound particles as opposed to 38% for pAb R8 and 89% as opposed to 14% for mAb MC23.

These results suggest either that cell binding triggered a reorganization of gD at the surface of cell-bound virions, or that the subpopulation of virions with ring-like distribution of gD is more competent for cell binding.

## gD localization at the surface of virions and capsid-less L-particles is comparable

Viral particles purified from cell supernatant often include capsid-less, non-infectious light particles (L-particles) as well as virions [17]. The composition of the envelope proteins of these

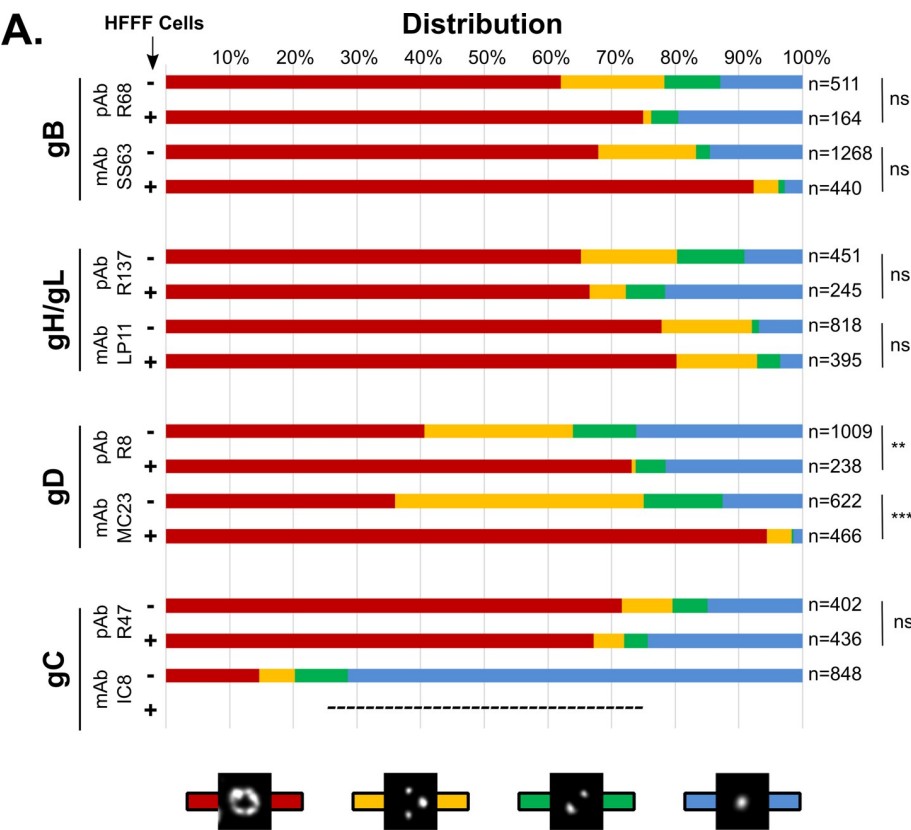

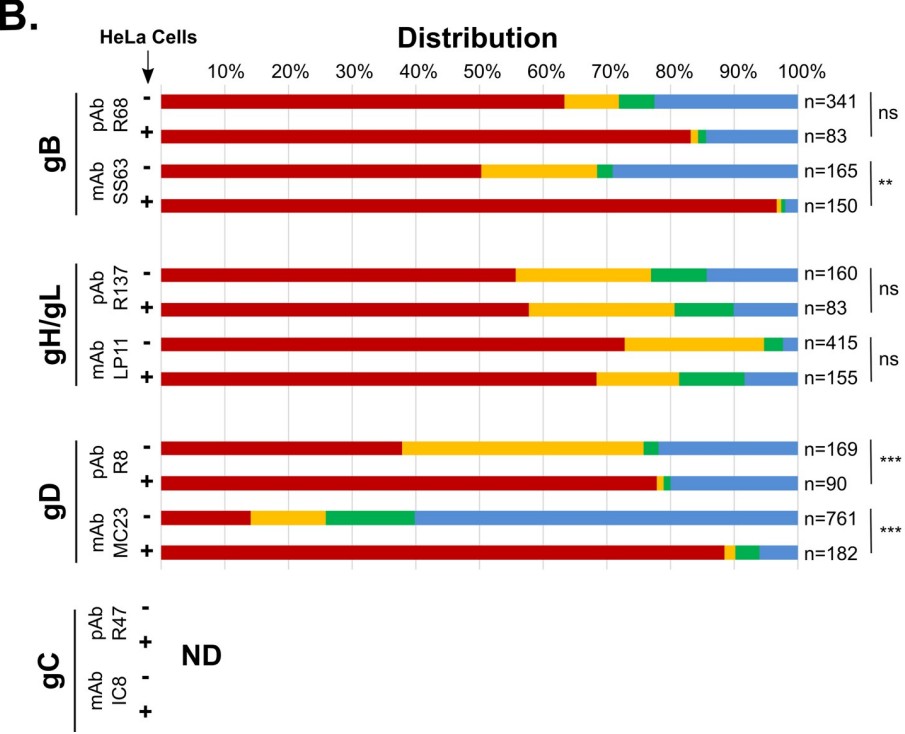

**Fig 2. Free and cell-bound particles may display different glycoprotein patterns.** Particles bound to cells (+) or attached to coverslips (-),were imaged using gSTED and the localizations of glycoproteins gB, gH/gL, gD and gC were

assessed by using a monoclonal (mAb) or a polyclonal antibody (pAb) recognizing each glycoprotein. Localization was categorized as single (blue), double (green) or multiple spots (yellow) or rings (red), as described in Fig 1. (A) Cells used for attachment were HFFF cells. Particles were attached to glass coverslips at room temperature whereas particles were bound to cells at 4°C to prevent membrane fusion. No IC8-positive signal (gC) could be detected on cell-bound particles. (B) Cells used for attachment were HeLa cells. Particles were attached to glass coverslips or bound to cells at 4°C. Localization of gC was not analyzed in this experiment. A Pearson's chi-squared test was used to determine whether the profile of distribution of one glycoprotein was statistically different between free and cell-bound virions. The p-value indicates the likelihood of a correlation, therefore a p-value > 0.05 was considered as indicating a statistically significant difference between the two sets. ns: $p<0.05$, **: $p>0.1$, ***: $p > 0.5$. n: number of analyzed particles per condition. ND: not determined.

particles is remarkably similar to that of virions and they are capable of membrane fusion and of delivering of some tegument proteins [18, 19]. A possible explanation for the difference of localization of gD between free and cell-bound particles could be that if L-particles have a distinctive organization of gD (rings or spots) these might be favored or lost during cell attachment. Although virions and capsid-less L-particles were separated by banding on Ficoll gradients, there is inevitable cross-contamination. Therefore, the purity was assessed by co-staining of permeabilized virions with antibodies specific for the capsid and for gD (S2 Fig). This showed that 73% of purified particles were positive for both capsids and gD, thus representing virions (S2B Fig) and that L-particles (positive for gD only) accounted for 24% of the total. To determine how much the presence of the remaining L-particles might interfere with the interpretation of glycoprotein distribution, the localization of gD was compared between cell-free virions and L-particles using the R8 polyclonal antibody against gD (S2C Fig). Although the distribution pattern of gD on L-particles was statistically different to that of virions, the numbers observed (36% of rings on virions and 25% on L-particles) were in the same range as those observed on total cell-free particles with the same antibody (39% in average) and different from the numbers observed on cell-bound particles (73% of rings on HFFF cells and 78% on HeLa cells).

Thus, the major increase in the proportion of particles with ring-like gD distributions observed on bound cells is unlikely to be linked to a selection of ring-enriched or a depletion of spot-enriched L-particles during cell binding.

## mAb IC8 recognizes an epitope of gC which is undetectable on cell-bound particles

Polyclonal antibody R47 against gC showed a distribution for the glycoprotein similar to the one observed with gB or gH/gL on free and cell-bound virions with 72% and 67% of rings respectively. However, labeling of gC with monoclonal antibody IC8 showed a strong spotty localization of the protein, essentially composed of a single spot on each virion (72%) (Fig 2A). We interpret this apparent contradiction to the detection by mAb IC8 of a subpopulation of gC that is localized at a specific position in the envelope, unlike the totality of gC molecules detected by pAb R47, which are spread evenly around the particle. This interpretation is supported by the failure of mAb IC8 to detect gC on cell-bound particles, in contrast to R47, possibly because this mAb recognizes an epitope of gC involved in receptor interaction.

## Rings of glycoproteins on cell-bound particles are smaller than those on free particles

Although no obvious change in the profile of localization was observed for gB and gH/gL, subjective comparison of images suggested that the ring diameters differed between free and cell-bound particles. To determine the extent of any differences, we measured the diameter of

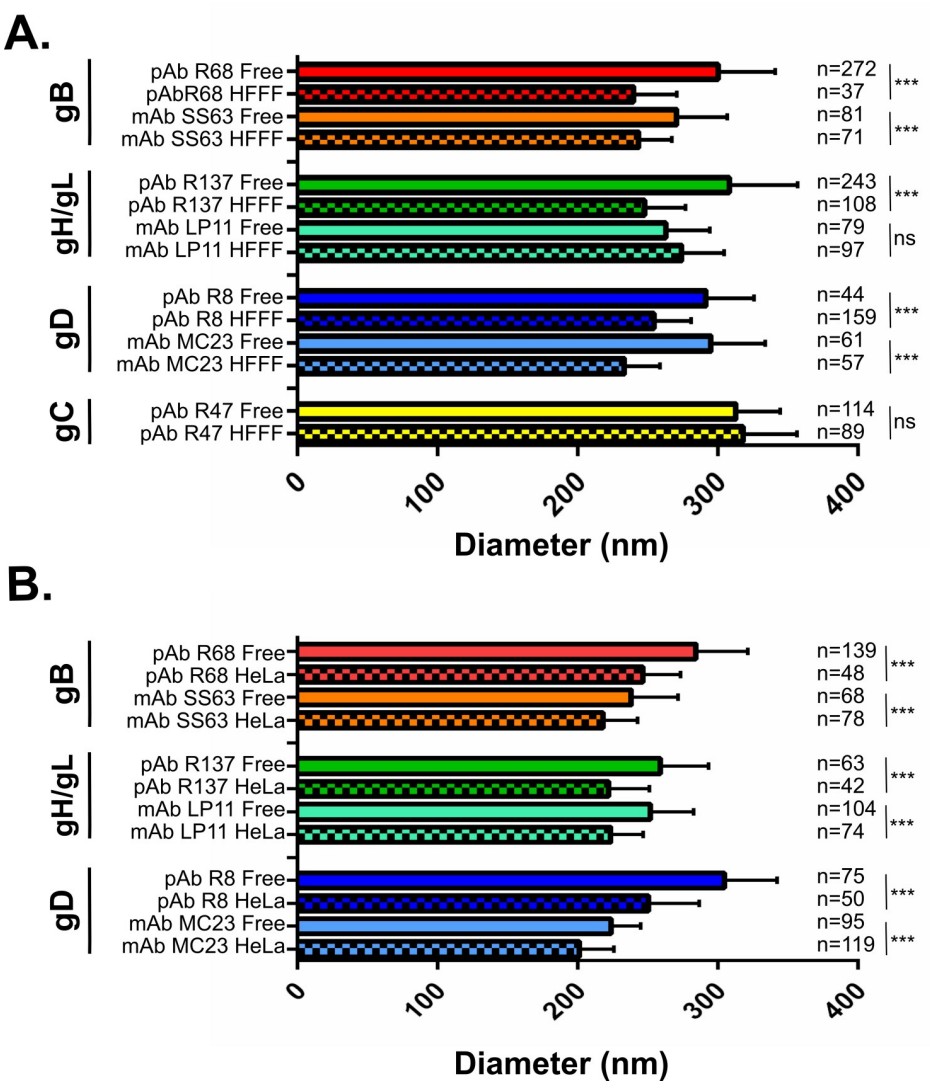

**Fig 3. The diameter of labeled rings may differ between glycoproteins and between free and cell-bound viral particles.** The diameters of 2,467 rings observed on raw, unmodified images obtained in gSTED mode were measured and compared according to the type of glycoprotein labeled and whether the particles were bound to cells or not. (A) Purified virions were attached to glass coverslips at room-temperature (Free) or bound to HFFF cells at 4˚C. (B) Purified virions were attached to glass coverslips at 4˚C (Free) or bound to HeLa cells at 4˚C. Statistical differences were measured using an unpaired student's t-test after the Gaussian distribution of values was verified using a Shapiro-Wilk normality test. ns: p >0.01, ***: p<0.0001 (B). n: number of analyzed particles per condition.

rings of different glycoproteins on free or cell-bound virions. The average diameters measured are shown in Fig 3 and representative pictures from the set of experiments shown in 3A with illustrative intensity profiles are shown in Fig 4. In the set of experiments shown in panel A, the average diameter of rings of free virions as shown by labeling with polyclonal antibodies was similar between gB, gH/gL and gD at 300, 307 and 291 nm respectively. Interestingly, there was a decrease of ~16% in the average ring diameter between free virions and cell-bound virions labeled with the same antibodies at 240, 248 and 254 nm respectively. Decreases were also observed with monoclonal antibodies against gB (-10%: from 270 nm to 243 nm) and gD (-21%: from 295 nm to 233 nm). There was no significant change in the ring diameter of gC between free and cell-bound particles with 313 and 318 nm respectively.

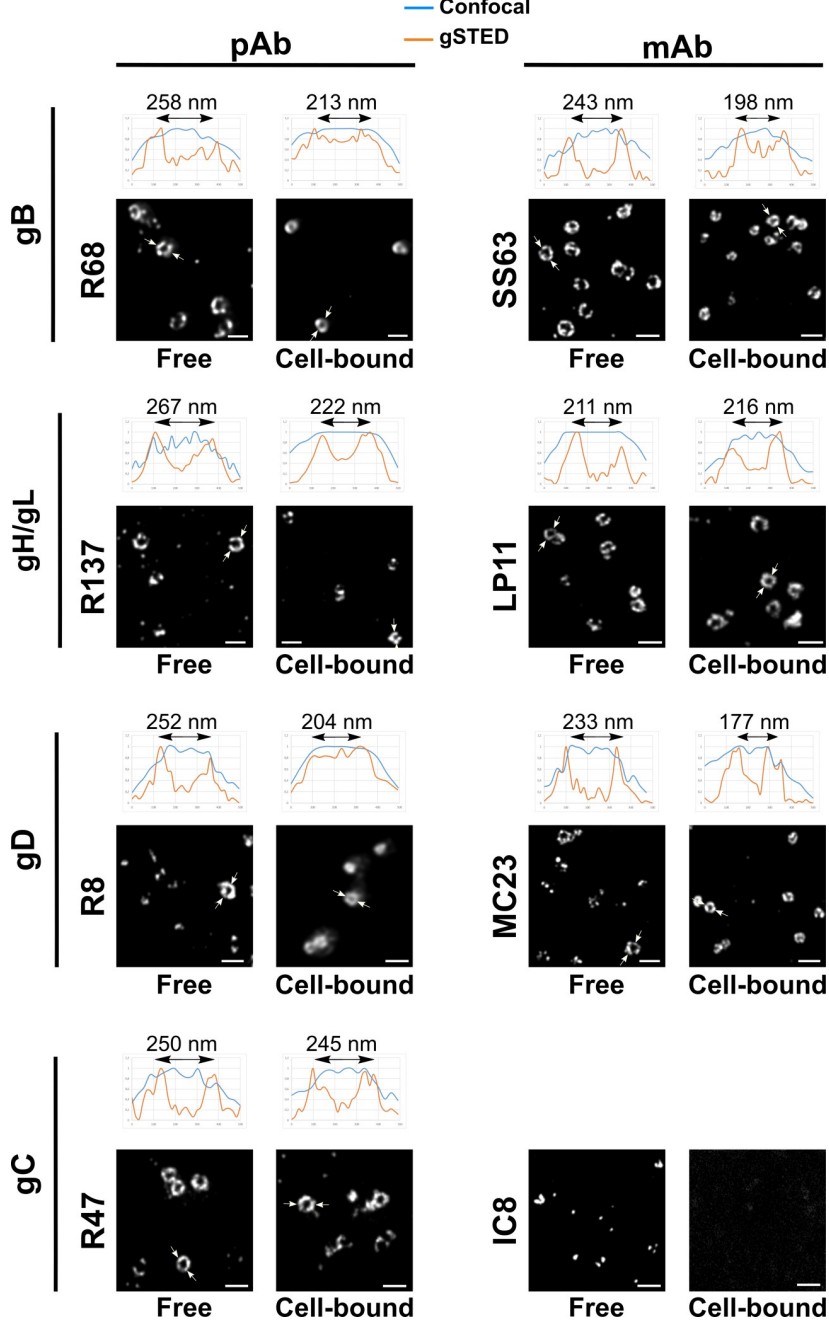

**Fig 4. Distribution of glycoproteins on cell-free on cell-bound viral particles.** Representative noise-filtered gSTED images of glycoprotein distribution on the surface of free or HFFF-bound viral particles for all available antibodies (see S3 Fig). The intensity profile of one particle per field along a 500 nm-long line delineated by two arrows is shown above each picture. Blue line: confocal profile; orange line: gSTED profile. The intensity profile was determined from the unmodified raw images. The peak-to-peak distance is indicated for each gSTED profile. No specific signal could be observed with the IC8 antibody on cell-bound particles. Scale bars: 500 nm.

In the set of experiments shown in panel B, the observed diameters of both free and cell-bound virions were 9% smaller on average than for those in the panel A essentially because of the higher resolution achieved here (44 nm as opposed to 60 nm, S1 Fig). Nevertheless, the

diameter of rings on HeLa-bound particles was here again consistently smaller than the diameter of rings on cell-free particles with an average decrease of 13%. This shows that the diameter of rings is lower on cell-bound particles than on cell-free particles and that this change is not linked to the cell type or to the temperature of incubation.

## Dual labeling efficiency of glycoproteins on free particles

In order to determine the relative localization of the glycoproteins, we carried out dual-color gSTED microscopy using the same sets of antibodies as above and combining a polyclonal antibody directed against one glycoprotein with a monoclonal antibody against a second.

For each pair of antibodies, we first quantified the number of free particles that were singly or doubly labeled (Fig 5A). In most cases, when labeling for gC was excluded, more than 90% of particles were dual-labeled. In the case of gC-labeled particles, 22% to 34% of particles stained with a polyclonal antibody against gD, gB or gH/gL and with mAb IC8 against gC had no gC signal (lanes x-xii, red bars). These numbers were different when dual-labeling was reverse (lanes i, vii and v, green bars). For instance, dual-labeling with IC8 and R68 (gB) showed that 34% of particles were gB+/gC- (lane xi, red bar), whereas dual-labeling with pAb R47 (gC) and mAb SS63 (gB) showed that only 8% of particles were gB+/gC- (lane vii, green bar). This illustrates that IC8 does not label all gC+ particles. Moreover, labeling using gC-specific polyclonal antibody R47 showed that 14 and 18% of particles were labeled for gC only and not for gB or gH/gL respectively (lanes vii and v, red bars). This was even more significant for particles labeled with gD mAb MC23 since 36% of them were gC+ and gD- (lane i, red bar). These findings were not dependent on the antibody used as when gC was labeled using mAb IC8 and the other glycoproteins were labeled with their respective polyclonal antibody, numbers of particles labeled with gC only were similar: 12% (gC+/gB-, lane xi, green bar), 16% (gC+/gH/gL-, lane xii) and 30% (gC+/gD-, lane x). This suggests that a significant number of particles (roughly around 15%), which are presumably defective, contain gC but none of the other glycoproteins tested. Notably, the fraction of particles that labeled only for gC was twice as great in gD co-labeled particles as for other glycoproteins (red bar in lane i and green bar in lane x). If 15% of particles contain only gC, this means that the remaining 15% of gC+/gD- particles are gB+/gH-gL+/gD-. However, we only observed 4.3% of gH-gL+/gD- particles (lane iv, green bar) and 5.5% of gB+/gD- (lane ix, green bar) when gD was labeled with pAb R8. Therefore, the most likely explanation for the existence of the high number of gC+/gD- particles is that co-labeling of gC and gD with their respective antibodies is mutually inefficient, possibly because the two molecules are very close together [20].

## Receptor-binding gD and gC and fusion-triggering gH/gL and gB have distinct localization on free particles

We then analyzed the localization in free virions of each glycoprotein relative to the others (Fig 5B). As expected, gB and gH/gL were found predominantly as complete rings, which resulted in overlapping signals (lanes vi and viii). Moreover, when single spots of both glycoproteins were found, they were often in close proximity to each other, indicating that the two proteins are close together on the viral envelope. gD was more frequently found as discrete spots rather than as rings and, surprisingly, these spots rarely overlapped with rings of gB or gH/gL and were often found in areas where the gB or gH/gL signals were the weakest (lanes ii, iii, iv and ix). We made similar observations for epitope IC8 of gC which localizes overwhelmingly as spots (lanes xi and xii). In contrast, spots of gD and gC were often observed as adjacent or colocalizing (lane i). Interestingly, we observed that although gC labeled with pAb R47 localized essentially as rings on mono-labeled particles, dual-labeled particles displayed

**A.**

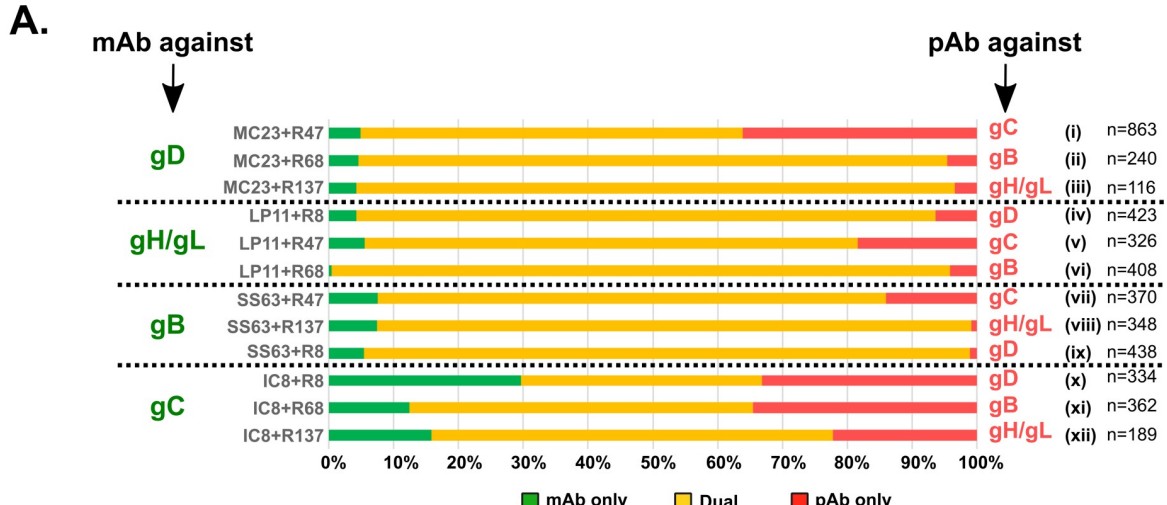

**B.**

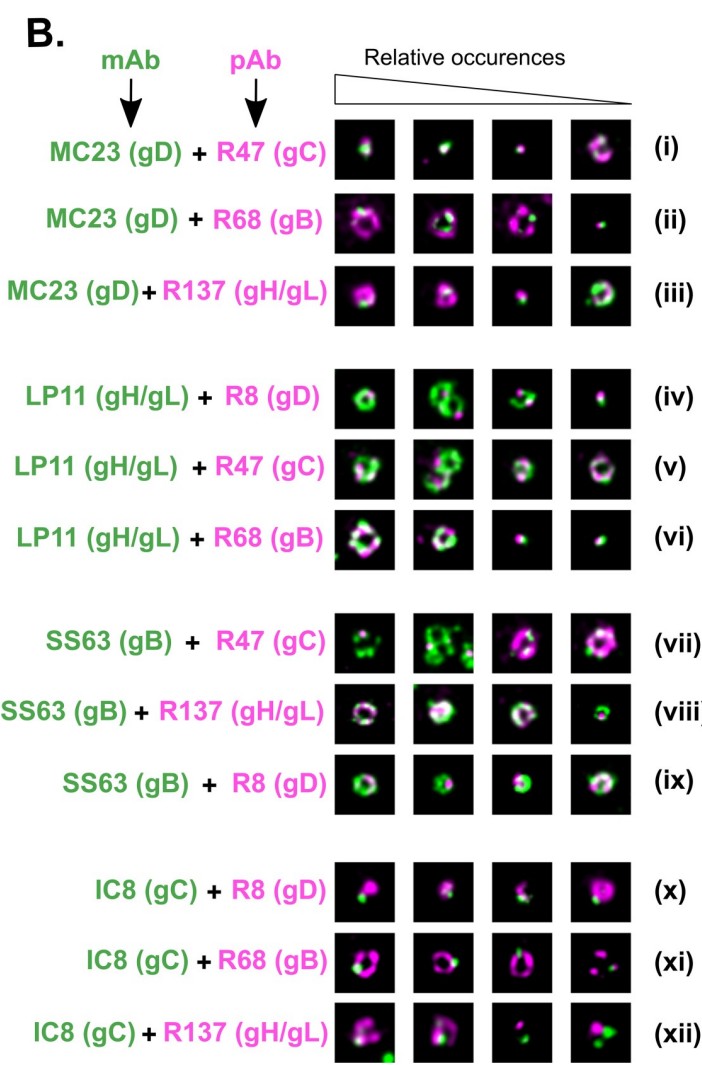

**Fig 5. Relative positions of glycoproteins on free virus particles as revealed by dual-color gSTED.** (A) Purified free particles were double-labeled with twelve different pairs of antibodies. In each case, the percentage of particles labeled with mAb only is shown in green, the percentage of particles labeled with pAb only is shown in red and the percentage of double-labeled particles is shown in yellow. n: number of analyzed particles per condition. (B) For every pair of antibodies, representative noise-filtered gSTED pictures of cell-free particles are shown. The images on the left show more commonly observed patterns while those on the right are less commonly observed. mAb staining is pseudo-colored in green and pAb staining is pseudo-colored in magenta. All pictures are 700 x 700 nm.

essentially spots of gC which, here again, did not colocalize with the rings of gB or gH/gL (lanes v and vii). In those occurrences where rings of gC were observed, gB and gH/gL where localized mostly as spots which rarely overlapped with the ring of gC.

We conclude that gB and gH/gL are more likely to be localized close together than either is to gD or gC. Similarly, a fraction of gD molecules seem to be sufficiently close to gC to be observed as adjacent and sometimes colocalizing spots in our set-up.

### Comparative dual-color STED microscopy on gD-positive free and cell-bound particles

Since the localization of gD is different between cell-free and cell-bound particles and gD has a localization distinct to gB and gH/gL on cell-free virions, we compared the patterns of glycoprotein distribution between cell-free and cell-bound particles on gD-positive particles in order to verify whether a dominant profile emerged.

First we compared the efficiency of dual-labeling on HeLa-bound particles to that of free particles. As shown in Fig 6A, dual-labeling was overwhelmingly efficient with almost all tested combinations of antibodies with between 80% and 92% of particles being dual-labeled. The only exception was co-labeling with mAb MC23 against gD and pAb R47 against gC. The majority of mono-labeled cell-free particles consisted of gC-labeled particles (28%) where gC was essentially localized as clusters (Fig 6C, lane i, grey bar). In contrast, mono-labeled cell-bound particles consisted mainly of particles labeled only for gD (50%) while particles labeled only for gC dropped to 4%. These observations indicate that the clusters of gC observed on cell-free particles are not visible on cell-bound particles, either because these particles are defective and/or not competent for cellular binding, or because those clusters are engaged in receptor-binding and are therefore inaccessible to the antibodies.

Next, we compared the distribution of each pair of glycoproteins tested on cell-free and cell-bound particles in order to determine if a particular distribution profile predominates (Fig 6B and 6C). This showed that, with the exception of gC, the distribution of glycoproteins at the surface of cell-bound particles was essentially composed of rings of glycoproteins (Fig 6C, red bars in lanes iv, vi, viii and x) that fully or partially overlapped (Fig 6B). This contrasts with free particles which were composed of a majority of rings of gB or gH/gL with spots of gD which were excluded from gB or gH/gL rings (Fig 6C, yellow bars in lanes iii and v and orange bars in lanes vii and ix and Fig 5B, lanes ii, iv and ix).

These results show that on cell-bound particles, all glycoproteins other than gC converge to produce a tighter more uniform distribution around the viral particle than is seen in free particles.

## Discussion

It was first established by cryo-electron tomography (cryo-ET) that the virion of HSV-1 has an asymmetrical architecture with the capsid close to the viral envelope at one pole and separated from the opposite side of the envelope by ~35 nm of tegument [12]. This intriguing feature has been confirmed by other groups using either conventional electron microscopy [21], cryo-ET

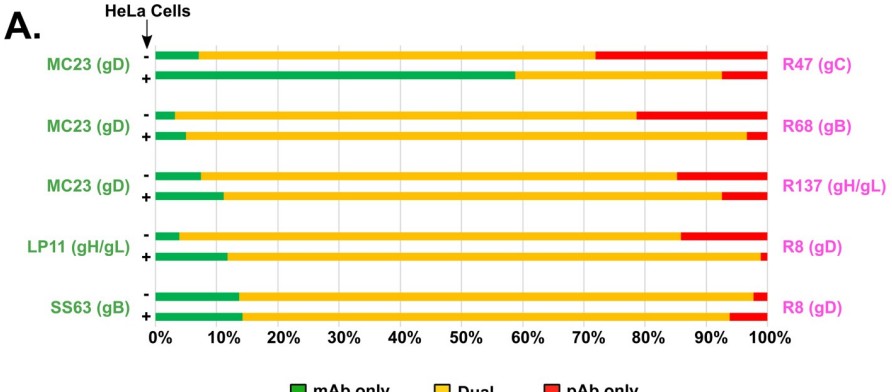

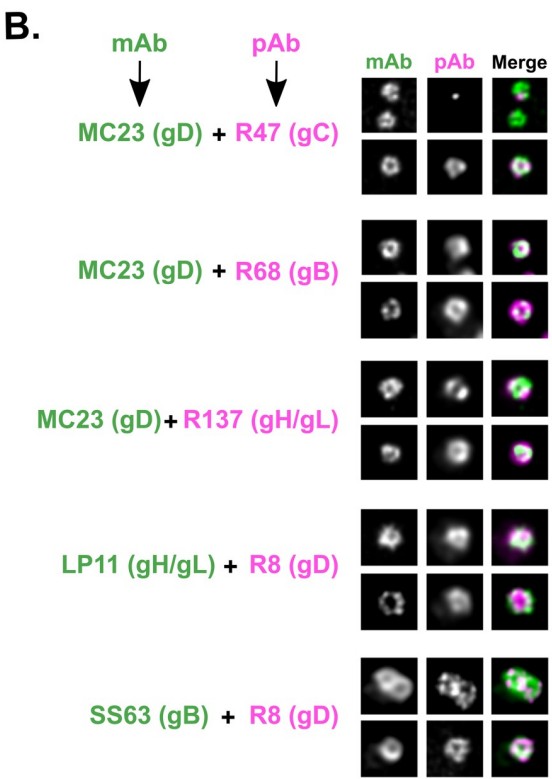

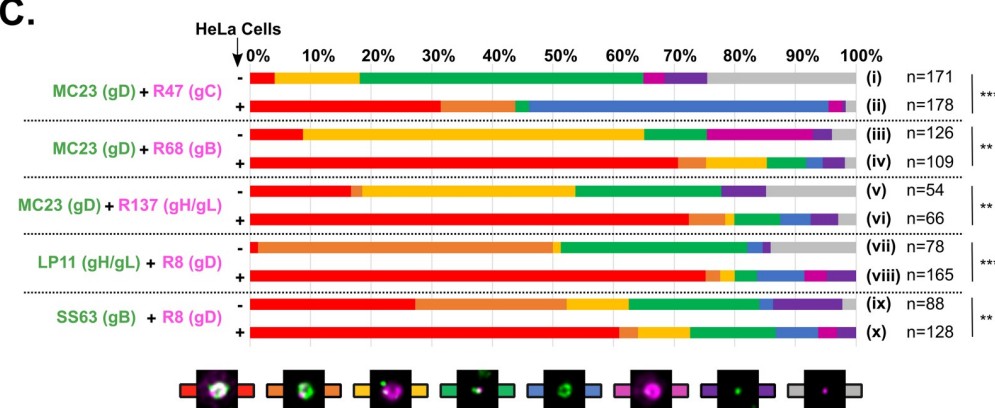

**Fig 6. Positions of glycoproteins relative to gD on HeLa-bound virus particles as revealed by dual-color gSTED.**
(A) Viral particles were attached to glass coverslips or bound to HeLa cells at 4°C and labeled with five different pairs of antibodies. In each case, the percentage of particles labeled with mAb only is shown in green, the percentage of particles labeled with pAb only is shown in red and the percentage of double-labeled particles is shown in yellow. (B) For every pair of antibodies, two different sets of representative noise-filtered gSTED pictures of HeLa-bound particles are shown. mAb staining is pseudo-colored in green and pAb staining is pseudo-colored in magenta. All pictures are 700 x 700 nm. (C) The distribution of two glycoproteins per condition was determined on free particles (-) or on particles bound to HeLa cells (+) and the different combinations were categorized into eight different profiles. The profiles used here illustrate combinations such as "ring-ring" (red bars), "ring-spot(s)" (orange bar), "spot(s)-ring" (yellow bar), "spot(s)-spot(s)" (green bar), "ring-no signal" (blue bar), "no signal-ring" (pink bar), "spot(s)-no signal" (purple bar) and "no signal-spot(s)" (grey bar) for "mAb-pAb" signals (colored in green and pink respectively). Single spots are shown here for illustrative purposes but spots can also be multiple. A Pearson's chi-squared test was used to determine whether the profile of distribution of one glycoprotein was statistically different between free and cell-bound virions. A low p-value indicates the likelihood of a correlation, therefore a p-value > 0.05 was considered as indicating a statistically significant difference between the two sets. ns: $p<0.05$, **: $p>0.1$, ***: $p > 0.5$. n: number of analyzed particles per condition.

with no imposed symmetry [22] or ensemble mapping of fluorescently tagged viruses [23]. The polarity created by this structural asymmetry is essentially visible through the eccentricity of the capsid within the particle and the differing density of the tegument between the two poles. Moreover, observation of viral progeny in the form of cell-associated virus by electron microcopy showed that these particles had a symmetrically arranged tegument, as opposed to purified, cell-free virus [21]. The tegument can undergo structural rearrangement, as was reported for the tegument protein pUL16 upon cell binding [24]. The observation of differing densities of glycoprotein spikes at the surface of viral particles suggests that glycoprotein distribution is also uneven [12]. Since numerous interactions exist between the tegument and glycoproteins [25–28], it is likely that tegument rearrangement is correlated with the localization of viral envelope proteins. Taken together, these observations argue for a dynamic nature of the tegument and some envelope proteins over time and/or upon cell attachment [29].

Here, using super-resolution fluorescence microscopy, we made two observations which relate to the organization of glycoproteins and their possible dynamics at the surface of viral particles.

First is the apparent "constriction" of rings of gB, gH/gL and gD upon cell-attachment. This change of diameter was reproducible and was quantified to around 13 to 16% on average (Fig 3). We exclude the possibility that this is caused by a general change of diameter of the whole viral particle since the diameter of gC rings remained unchanged upon cell binding. Alternatively, change of glycoprotein conformation could lead to an overall decrease in ring diameter, although this is unlikely because (i) all glycoproteins except gC should then undergo conformational change and (ii) the size of gD, gH/gL and the pre-fusion form of gB have been shown or estimated to be 6 nm, 10 nm and 10 nm long respectively [13, 30–32]. This means that a change of conformation from a 10nm-long molecule to roughly half this size would result in a reduction of ring diameter of 10 nm in general (two times 5 nm), which is much less than what was experimentally observed (40 nm in average). Instead, we suggest that this constriction could represent a regrouping of glycoproteins at one pole of the virion. Rings are considered to represent a confocal slice of virions where the glycoproteins would be evenly distributed around the particle (STED do not resolve in z with our set-up). In the case where proteins would be regrouped, the diameter of rings would be expected to decrease as the regrouping increases.

Interestingly, observation by cryo-ET of a single cell-bound HSV-1 particle suggested that it was bound to the cell through the capsid-proximal pole of the viral envelope, which had a lower density of glycoprotein spikes than the opposite pole (or distal pole) [12, 33]. However, the proteins regrouped on the side of the plasma membrane and those regrouped on the

opposite side could not be identified. As suggested by Maurer et al., it is possible that glycoproteins involved in cell binding and fusion, such as those analyzed in our study, regroup locally at the point of contact with the cell while others, involved in the negative regulation of some of these processes such as gM [34, 35] or involved in the modulation of the immune response are concentrated on the opposite side.

Although the change in ring diameters of gB, gH/gL or gD upon cell attachment is relatively obvious (Fig 3), the marked difference between the distribution pattern of gD at the surface of free virions and that of cell-bound virions provides probably the best evidence that the localization of glycoproteins at the viral envelope is dynamic. In the first set of experiments, we observed that only 40% of free particles have a ring-like distribution of gD when labeled with pAb R8 (36% with mAb MC23), as opposed to >60% for other glycoproteins. These results were confirmed in a second set of experiments where cell-free particles were incubated at 4°C rather than room temperature, with 38% of ring-like gD with pAb R8 and no more than 14% with mAb MC23. We believe that since multiple spots were composed in a large majority of three or four spots, they represent a stable state of clustered glycoproteins. In contrast, particles with five or more spots were very rare, yet visible and distinct from rings. Therefore they could represent an intermediate state between "spots" and "rings" where glycoproteins are in the process of reorganizing around the viral particle.

We excluded the possibility that the presence of spots rather than rings was due to under-labeling of samples because all antibodies, representing multiple epitopes within the same glycoprotein, were used at the same dilution. In conditions of under-labeling, antibodies with the lowest concentration would be expected to give more single spots than antibodies which are the most concentrated. As illustrated in S3 Fig, this was not what was observed, since (i) the most concentrated antibody (IC8, 8.6 mg/mL) was the antibody giving the largest number of particles with single spots (72%, see Fig 3); (ii) gD antibodies MC23 and R8 gave similar pattern of localization despite the fact that R8 is 5.4 times more concentrated than MC23 and (iii) MC23 showed 94% of rings on cell-bound viral particles despite being the least concentrated antibody of our stock (1.2 mg/mL). An additional argument for the specificity of the single or multiple spots of gD comes from dual-labeling of particles which showed that at least 90% of particles labeled with either antibody against gD were also co-labeled for another glycoprotein (gC excepted) (Fig 5).

We observed this clustered localization of gD on different viral preparations, with two different antibodies and on a large number of particles. Yet, it was not reported in previous studies of the viral structure at high resolution. Using STORM (Stochastic Optical Reconstruction Microscopy), another method of super-resolution microscopy, Laine and collaborators reported on the ring-like distribution of gD on purified, permeabilized particles using mAb LP2 [36], which binds an epitope very close to the one recognized by MC23 used in our study [37, 38]. They did not report gD as being present in single spots possibly because the authors assumed a spherical distribution of gD from the start, and focused on determining the parameters of those spheres in a model-imposed reconstruction. Using diffraction-limited fluorescence microscopy, Bohannon et al. used an elegant system where the centroid of fluorescence was determined by 2D Gaussian fitting to model the relative position of fluorescently tagged proteins of recombinant pseudorabies virus (PrV) virions [23]. This showed that gD localization on free virions was not eccentric on the basis that the average displacement of fluorescence from the capsid was less or equal to the one expected from the asymmetric organization of the particle (15 +/- 5 nm). In the case of the double or multiple spots of gD that we observed, the average displacement would be expected to be low as well when spots are radially distributed from the center of the particle (which was observed frequently as in Figs 1 and 4) and thus have a barycenter close to the one of a whole envelope distribution of gD. Considering

that the total of free particles we observed with double or multiple spots or rings of gD exceeds 70% of the cell-free particles (except for MC23 in experiment set #2, Fig 2), our results are compatible with those of Bohannon et al. Other possible explanations include (i) the absence of clusters of gD on PrV envelopes as opposed to HSV-1; (ii) the GFP fusion on gD altering its distribution and/or (iii) the transition of gD from the clustered state to the evenly distributed state (as described below) is faster on PrV than on HSV-1 envelopes.

There are several possible scenarios which can account for the difference of localization of gD at the surface of free and cell-bound particles. In contrast to particles with gD all around the envelope, particles with clusters of gD could be defective or immature particles which cannot bind to cells and are therefore lost in our cell-binding assays. Such immature particles could undergo a structural change resulting in gD being redistributed all around the particle. This change could be triggered by the first steps of binding to the cell (such as the interaction of gC or gB with heparan sulfates) and/or could occur naturally with time [21]. A similar pattern of change of glycoprotein distribution upon particle maturation is seen in HIV-1 [39]. Using STED microscopy, the authors observed that Env proteins were clustered at the surface of mature viral particles whereas immature particles displayed glycoproteins which were organized as multiple spots or in a ring-shape. Although the clustering of gD resembles that of Env, it seems to have opposite functions. Clustered Env is associated with mature HIV-1 particles whereas it is the non-clustered form of gD which is associated with cell-bound HSV-1.

The reorganization of Env at the surface of viral particles was shown to be triggered by the cleavage of the env-associated matrix protein from the capsid which primes the particle into a mature state[39]. In the case of HSV-1, it is conceivable that a similar mechanism exists. If it does, it could be through a mechanism of "inside-out" signaling or "outside-in". In the "inside-out" scenario, changes in the tegument would translate into a change of localization of some glycoproteins such as gD making the viral particle mature, i.e competent for cell binding, in the manner of what was described with HIV-1. In the "outside-in" scenario, an event associated with adsorption of the viral particle onto the cellular membrane such as binding of gC or gB to heparan sulfates could trigger a signal in the tegument that would lead to tegument rearrangement and the subsequent relocalization of some glycoproteins such as gD.

It is unclear how gD redistribution around the viral particle would make it more competent for cell binding. One hypothesis is that since binding of gD to its receptors is necessary to activate gH/gL and gB for membrane fusion to occur [40, 41], gD could be clustered on the immature particle into non-functional or inactivating domains within the viral envelope that inhibit its possible association with receptors. Upon maturation, gD would be freed from these clusters and be available to perform its functions. In addition to the core glycoproteins described here, herpesviruses contain ten different "non-essential" proteins at their surface (in addition to gC) with unknown or unclear functions that could be candidates for having such an inhibitory role [42].

In conclusion, our results shed new light on specific glycoprotein distribution at the surface of viral particles showing that this distribution is dynamic and that different populations of a given protein may exist within the particle, possibly with differing roles. Super-resolution fluorescence microscopy in combination with cryo-ET are thus expected to contribute significantly to our understanding of the in-depth composition of large viruses and the dynamics of their structures.

## Methods

### Cells and viruses

HeLa cells (obtained from the American Type Culture Collection, ATCC) and Human Fetal Foreskin Fibroblasts (HFFF2, obtained from the European Collection of Authenticated Cell

Cultures, ECACC) were grown at 37˚C in Dulbecco's modified Eagle medium (DMEM; Gibco) supplemented with 8% fetal calf serum.

The HSV-1 strain used in this study is the 17+ strain. High-quality virions were generously provided by Marion McElwee and Frazer Rixon (University of Glasgow, UK). Virus was propagated in Baby Hamster Kidney cells (BHK, ATCC) grown in Glasgow MEM (GMEM), 10% fetal calf serum and 10% tryptose phosphate broth. Confluent cells were infected at a m.o.i. of 0.002 pfu/cell and were incubated at 37˚C for three days. Cells were then harvested by shaking into the medium. Cells and media were centrifuged at 1600× g for 10 minutes to remove cell debris. Virus was pelleted by centrifugation at 17,000× g for two hours. The pellet was gently resuspended on ice overnight by overlaying with 2 ml GMEM. The resuspended material was moved to a new tube and clarified by spinning at 200× g for ten minutes. The supernatant was layered onto the top of a 5%-15% Ficoll gradient prepared in GMEM and centrifuged at 26,000× g for 2 hours. The upper band containing the virions was collected by side puncture of the tube using an 18-gauge needle. Collected virions were diluted in GMEM and pelleted at 40,000× g for 1 hour. The pellet was washed gently in PBS and then allowed to resuspend in 50–100 μl PBS by incubating on ice for at least one hour.

Four different preparations of virions were used for this study.

## Antibodies

All monoclonal and polyclonal antibodies against viral glycoproteins used in this study are listed in S3 Fig and were described elsewhere [20, 30, 38, 43–50]. The PTNC antibody was raised against purified nuclear capsids and was described previously [51]. Secondary antibodies used were goat anti-mouse or goat anti-rabbit conjugated with either Oregon Green 488 or AlexaFluor 532 (ThermoScientific) for gSTED microscopy and AlexaFluor 488 and AlexaFluor 555 (ThermoScientific) for analysis of the viral preparations by conventional fluorescence microscopy.

## Sample preparation for gSTED microscopy

A suspension of purified WT 17+ virions was layered on type #1.5H coverslips with thickness of 0.170 +/-0.005 mm (Zeiss) in 6-well plates and incubated for 30 minutes either at RT (first set of experiments) or at 4˚C (second set of experiments). Coverslips were washed three times with PBS and fixed with 4% paraformaldehyde. Samples were incubated with 0.2% fish gelatin (Sigma) and 5% goat serum to saturate non-specific binding sites. Glycoprotein labeling was carried out by incubating the fixed particles with primary antibodies at a dilution of 1/100 for 1h at RT. Secondary labeling was performed by incubating the samples with secondary antibodies at a dilution of 1/50 for 1h at RT followed by three washes in PBS. Samples were then mounted with Prolong Gold (Life Technologies) and cured before imaging. High concentrations of antibodies were necessary to obtain a signal strong enough for gSTED imaging and to ensure saturation of all available epitopes.

For observation of cell-bound virions, HeLa or HFFF2 cells were seeded on sterile type #1.5H coverslips with thickness of 0.170 +/-0.005 mm (Zeiss) 24h before incubation with 50 pfu (plaque-forming units) per cell of WT virions for 1h at 4˚C. Unbound virus was washed off with PBS (three washes) and cells were fixed and processed as described above.

## gSTED microscopy and noise-filtering

gSTED microscopy was carried out on a Leica TCS SP8 confocal microscope equipped with a white light laser (WLL operating at 70% of its nominal power), a hybrid detector, a 592 nm depletion laser and a gating system. Observations were made using a HCX PL APO 100X oil-

immersion objective (N.A = 1.4). Visualization of Oregon Green 488-labeled particles was done with an excitation wavelength of 495 nm at a power of 8%. The detection window of the Acousto-Optical Beam Splitter (AOBS) was set at 508–570 nm with a time-gated detection window of 2 to 6 ns. Detection was using a Hybrid Detector (HyD) with an unmodified gain of 100%. Depletion was obtained with a power ranging from 70 to 80% on the 592 nm depletion laser. Visualization of AlexaFluor 532-labeled particles was with an excitation wavelength of 532 nm at a power of 20%. The detection window of the AOBS was set at 548–573 nm with a gated detection window of 1 to 4.5 ns. Detection was using a Hybrid detector with a reduced gain of 61%. Depletion was obtained with a power of 15% on the 592 nm depletion laser. The same settings were applied to all samples. This could result in a less effective depletion of diffracted light in samples with a stronger signal, such as those obtained with pAbs. Thus, ring diameters obtained with pAbs can occasionally be larger than those obtained with mAbs (Fig 3).

Noise-filtering was done using Huygens software (Scientific Volume Imaging) with the following parameters: an excitation fill factor of 1.2, a saturation factor of 45 and an immunity fraction set to 1%. Background level was defined according to the quality of each image independently. Noise-filtered pictures were used only for illustration purposes and not for quantifications.

## Quantification and statistics

All quantifications were performed on raw images to avoid any potential modifications introduced by the noise-filtering procedure. Dual labeling was assessed using Leica LAS AF lite software.

Glycoprotein distribution was quantified manually using ImageJ software on a total of 11,067 (Fig 2) and 1,033 particles (Fig 6C). A Pearson's chi-squared test was used to determine whether the profiles of distribution were statistically different between free and cell-bound virions. As the p-value indicates the likelihood of a correlation, a p-value $> 0.05$ was considered as representing a statistically significant difference. Ring diameters shown on Fig 3 were measured manually from edge to edge on 2,467 particles using Leica LAS AF lite software. The possible difference between the diameters of free and cell-bound viruses was tested by a unpaired student's *t* test with a significance threshold set at $p < 0.01$ (significance level: 1%), after the Gaussian distribution of the values was verified by a Shapiro-Wilk test for $p > 0.05$.

## Supporting information

**S1 Fig. Determination of the gSTED resolution by FWHM analysis.** (A) Free virions were attached to glass coverslips and incubated with mAb IC8 against gC or irrelevant anti-GFP monoclonal antibodies. In addition, uninfected HeLa cells were incubated with pAb R8 against gD. All samples were incubated with Oregon-green 488-conjugated secondary antibodies. The nonspecific signal consisting essentially of immune complexes was then imaged using the diffraction limited confocal mode, or the gSTED set-up using the same conditions as those described for imaging of glycoproteins. Scale bar: 2 μm. (B) Enlargement of the regions boxed in A and the corresponding intensity profiles shown along a line of 400 nm. Scale bars: 200 nm. To determine the resolution of the gSTED set-up, the full-width at half maximum (FWHM) was calculated for six different images per set of experiments. One is illustrated here for each set. The average of FWHM was 60 nm for the first set of experiments (52 nm shown here) and 44 nm for the second set of experiments (39 nm here).
(TIF)

**S2 Fig. Influence of L-particle contamination on the localization of gD in cell-bound virions.** (A) Preparations of purified virus particles were attached to glass coverslips at room-temperature, fixed, permeabilised and labeled with antibody MC23 against gD (green) and

antibody PTNC against capsids (red). Scale bars: 5 μm. (B) Quantification of the percentage of virions, L-particles and capsids in 17+ virion preparations. Virions were defined as particles positive for both capsid (PTNC) and gD (MC23) signals (yellow), L-particles (green) were defined as negative for capsid and positive for gD and isolated capsids (red) were defined as positive for capsid and negative for gD. (C) 17+ viral particles were banded on a Ficoll gradient to separate virions from L-particles. Particles were attached to glass coverslips at room-temperature and labeled with anti-gD polyclonal antibody R8. The distribution of gD according to the pattern defined in Fig 2 is shown. A Pearson's chi-squared test was used to determine whether the profile of distribution between virions and L-particles was significantly different. The p-value indicates the likelihood of a correlation, therefore a p-value > 0.05 was considered as indicating a statistically significant difference between the two sets. p = 0.23 (**).
(TIF)

**S3 Fig. Summary of all antibodies used in this study and the corresponding patterns of gly-coprotein distribution as described in Fig 2A.** Color-coding is identical as that of Fig 2: red: rings; yellow: multiple spots; green: double spots and blue: single spots. "Epitopes" indicates the residues or domains involved in antibody binding. References are listed in the Methods section. mar: mAb resistant mutation. (*) partial blocking of domains I (20%), II (15%) and IV (40%) of gB. (**) blocks several known epitopes of gD (residues 10–20, 67, 246, 75–79, 213 (MC23) and 262–279).
(TIF)

# Acknowledgments

We are indebted to Frazer Rixon and Marion McElwee (University of Glasgow, UK) for the gift of purified virions and for reviewing the manuscript. Many thanks to Eva Hernandez for her help with cellular preparations. We are grateful to Yves Gaudin for fruitful discussions and for critical reading of the manuscript. We acknowledge the help of the Département IBiSA des Microscopies of the University of Tours. We also thank Caroline Denesvre for her comments and for her support of this work.

# Author Contributions

**Conceptualization:** David Pasdeloup.

**Data curation:** Frauke Beilstein, David Pasdeloup.

**Formal analysis:** Frauke Beilstein, Audrey Esclatine, David Pasdeloup.

**Funding acquisition:** Audrey Esclatine.

**Investigation:** Frauke Beilstein, David Pasdeloup.

**Methodology:** Frauke Beilstein, Valérie Nicolas, David Pasdeloup.

**Project administration:** David Pasdeloup.

**Resources:** Gary H. Cohen, Roselyn J. Eisenberg.

**Supervision:** David Pasdeloup.

**Validation:** Frauke Beilstein, David Pasdeloup.

**Visualization:** Frauke Beilstein, David Pasdeloup.

**Writing – original draft:** David Pasdeloup.

**Writing – review & editing:** Frauke Beilstein, Gary H. Cohen, Valérie Nicolas, Audrey Esclatine, David Pasdeloup.

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
