## [Decision Letter · Decision Letter 0]

14 Oct 2019

Dear Dr Pasdeloup:

Thank you very much for submitting your manuscript "Dynamic organization of Herpesvirus glycoproteins on the viral envelope revealed by super-resolution microscopy" (PPATHOGENS-D-19-01587) for review by PLOS Pathogens. Your manuscript was fully evaluated at the editorial level and by independent peer reviewers. The reviewers appreciated the attention to an important topic but identified some aspects of the manuscript that should be improved.

We therefore ask you to modify the manuscript according to the review recommendations before we can consider your manuscript for acceptance. Your revisions should address the specific points made by each reviewer.

(1) A letter containing a detailed list of your responses to the review comments and a description of the changes you have made in the manuscript. Please note while forming your response, if your article is accepted, you may have the opportunity to make the peer review history publicly available. The record will include editor decision letters (with reviews) and your responses to reviewer comments. If eligible, we will contact you to opt in or out.

(2) Two versions of the manuscript: one with either highlights or tracked changes denoting where the text has been changed; the other a clean version (uploaded as the manuscript file).

We hope to receive your revised manuscript within 60 days or less. If you anticipate any delay in its return, we ask that you let us know the expected resubmission date by replying to this email.

[LINK]

Sincerely,

Yasuko Mori

Guest Editor

PLOS Pathogens

Erik Flemington

Section Editor

PLOS Pathogens

Kasturi Haldar

Editor-in-Chief

PLOS Pathogens

orcid.org/0000-0001-5065-158X

Grant McFadden

Editor-in-Chief

PLOS Pathogens

orcid.org/0000-0002-2556-3526

Reviewer's Responses to Questions

**Part I - Summary**

Reviewer #1: In this manuscript, Beilstein et al investigate the organization of several envelope glycoproteins (gps) of herpes simplex virus-1 (HSV-1) at the surface of cell-free and cell-attached viral particles. They use stimulated emission-depletion (STED) microscopy, a new tool which allows to achieving an average visual resolution of 60 nm, i.e. inferior to the size of viral particles (200nm), to visualize the sub-structural composition of the viral envelope. They chose to investigate 5 of the main envelope gps (gC, gD, gB and the gH-gL complex) implicated in cell entry using a panel of monoclonal and polyclonal antibodies, to explore their respective localization at the envelope surface, their colocalization, and putative redistribution due to cell attachment. To obviate any bias, they carefully compared results obtained with monoclonal and polyclonal antibodies, in particular when they analyzed colocalization of two gps, by combining a polyclonal antibody directed to one gp with a monoclonal antibody directed to the second, in both associations.

The main results suggest that gD localizes differently in cell-free and cell-attached particles; that gps functionally involved in entry, gB and gH-gL, overlap on cell-free particles; that all glycoproteins except gC converge and are dynamically redistributed around cell-bound particles.

The experiments are cautiously done and the results are precisely described and overall convincing, and provide new insights which should be of great interest to the scientific virology community. I however feel that several clarifications could improve the manuscript.

Reviewer #2: Stimulated emission depletion microscopy (STED) is a fluorescence microscopy technique that overcomes the diffraction limited resolution of confocal microscopes. The manuscript by Beilstein et al. explores herpes simplex virus 1 (HSV-1) glycoproteins on the purified virions through this new technique. They found that gB and gH/gL are distributed around purified virions., although the other essential glycoprotein gD localizes as clusters which are distinct from gB and gH/gL. Furthermore, they reported that all glycoproteins showed a similar ring-like pattern when virions were adsorbed to the cells, but the diameter of these rings was significantly smaller than those observed on cell-free viruses. Their results suggested that gD can be reorganized on the viral envelope following either a possible maturation of the viral particle or its adsorption to the cell. The authors suggested the model, in which the redistribution of glycoproteins during adsorption could contribute to initiate the cascade of activations leading to membrane fusion.

The article is well written, and clearly shows the unexpected character of viral glycoproteins during entry using super resolution microscopy. As it is not totally understood how viral glycoproteins on the virions induce membrane fusion during viral entry. New techniques such like super resolution microscopy and live imaging should shed light on viral entry, which is essential for viral replication. Thus, this article will be of broad interest in the readership. I have several suggestions for improving this manuscript.

**Part II – Major Issues: Key Experiments Required for Acceptance**

Reviewer #1: No supplemental experiment required. The authors should provide several additional statistical analyses (see below).

Reviewer #2: (i) Upon binding to the cells, the authors claimed the rings of glycoproteins were constricted. However, it is many possibilities in addition to the change on the position of glycoproteins. The authors should mention the other possibilities, such as global change of the modifications of epitopes on the glycoproteins.

(ii) Can the authors estimate the number (ratio) of gD among virions?

(iii) The authors should show the methods of collecting the virions. Such as cell lines, MOI or times after infection. These conditions effect composition of virions dramatically.

**Part III – Minor Issues: Editorial and Data Presentation Modifications**

Reviewer #1: 1. Have epitopes recognized by the different mAbs been identified, and if so can the authors provide their localization? This information could be helpful for interpretation of results (see lines 170-179, 231, and 347-348).

2. The authors report 4 different profiles of gp localization, but they mainly distinguish the “ring-like” as opposed to the other three profiles in their interpretation of results (Fig 1). First, do the authors suggest that the single, double and multiple profiles correspond to a similar functional state as opposed to the ring-like, or that each of these profiles correspond to different states/ functions? Second, were there non “ring-like” profiles containing more than 3 spots, or is 3 the largest distinguishable number of individualized spots in the multiple profile (i.e., how do the authors differentiate the multiple and “ring-like” profiles)? This point should be clarified since the whole discussion is based on this distinction : in particular, adding the “multiple” and the “ring-like” profiles would eliminate the distinction between the cell-free and the cell-attached localization of gD in HFF cells, and in HeLa cells analyzed with a pAb.

3. The results obtained using polyclonal and monoclonal antibodies are not strictly identical, mABs tending to provide more ring-like images on cell-bound particles than pAbs (see Fig 2 gB, gD). Are these differences significant and if so, how can they be explained? (see Q1).

4. A constriction of the diameter of rings of gB, gH-gL and gD is observed when comparing cell-free and cell-attached virions, ranging from 12 to 21% (Fig 3). There is also a difference in the diameter of gp rings on cell-free virions using pAbs compared with mAbs (Fig 3A, see gB and gH/gL, and Fig3B, see gB and gD): is it significant and how can it be explained?(see Q1).

5. Figure 5 : the interpretation that spots of gD and gC colocalize (lines 243-244, and 250-251) is not obvious (compare lanes x and xi).

6. Figure 6 C : The results show that gD is clustered at the surface of cell-free particles, and redistributed in a ring-like organization in cell-bound particles. As the authors remind (p 14 line 316), glycoproteins involved in cell binding and fusion should regroup locally at the point of contact with cells : indeed maturation of HIV particles is contemporary with the egress of mature particles from cells, and env clustering is supposed to render the virus productive for infection (Chojnacki, Nat Commun 2017). Therefore, the results shown in this manuscript seem to contradict previous results obtained in another model. Can the authors clarify further this point in their discussion ?

7. Since redistribution of gD was observed at 4°C, i.e. in conditions which permit attachment but preclude fusion, can the authors comment on glycoprotein mobility in membranes and in particular at that temperature?

PLOS authors have the option to publish the peer review history of their article (what does this mean?). If published, this will include your full peer review and any attached files.

Reviewer #1: No

Reviewer #2: No

---

## [Editor Report · Decision Letter 1]

13 Nov 2019

Dear Dr Pasdeloup,

We are pleased to inform that your manuscript, "Dynamic organization of Herpesvirus glycoproteins on the viral envelope revealed by super-resolution microscopy", has been editorially accepted for publication at PLOS Pathogens. 

Before your manuscript can be formally accepted and sent to production, you will need to complete our formatting changes, which you will receive by email within a week. Please note that your manuscript will not be scheduled for publication until you have made the required changes.

IMPORTANT NOTES

(1) Please note, once your paper is accepted, an uncorrected proof of your manuscript will be published online ahead of the final version, unless you’ve already opted out via the online submission form. If, for any reason, you do not want an earlier version of your manuscript published online or are unsure if you have already indicated as such, please let the journal staff know immediately at plospathogens@plos.org.

(2) Copyediting and Proofreading: The corresponding author will receive a typeset proof for review, to ensure errors have not been introduced during production. Please review the PDF proof of your manuscript carefully, as this is the last chance to correct any errors. Please note that major changes, or those which affect the scientific understanding of the work, will likely cause delays to the publication date of your manuscript. 

(3) Appropriate Figure Files: Please remove all name and figure # text from your figure files. Please also take this time to check that your figures are of high resolution, which will improve the readbility of your figures and help expedite your manuscript's publication. Please note that figures must have been originally created at 300dpi or higher. Do not manually increase the resolution of your files. For instructions on how to properly obtain high quality images, please review our Figure Guidelines, with examples at: http://journals.plos.org/plospathogens/s/figures.

(4) Striking Image: Please upload a striking still image to accompany your article if one is available (you can include a new image or an existing one from within your manuscript). Should your paper be accepted, this image will be considered for our monthly issue image and may also appear on our website to feature your article. Please upload this as a separate file, selecting "striking image" as the file type upon upload. Please also include a separate "Other" file with a caption, including credits and any potential copyright information. Please do not include the caption in the main article file. If your image is from someone other than yourself, please ensure that the artist has read and agreed to the terms and conditions of the Creative Commons Attribution License at http://journals.plos.org/plospathogens/s/content-license. Please note that PLOS cannot publish copyrighted images.

(5) Press Release or Related Media: If your institution or institutions have a press office, please notify them about your upcoming paper at this point, to enable them to help maximize its impact. If they will be preparing press materials for this manuscript, please inform our press team in advance at plospathogens@plos.org as soon as possible. We ask that you contact us within one week to plan ahead of our fast Production schedule. If you need to know your paper's publication date for related media purposes, you must coordinate with our press team, and your manuscript will remain under a strict press embargo until the publication date and time. This means an early version of your manuscript will not be published ahead of your final version. 

(6)  PLOS requires an ORCID iD for all corresponding authors on papers submitted after December 6th, 2016. Please ensure that you have an ORCID iD and that it is validated in Editorial Manager.  To do this, go to ‘Update my Information’ (in the upper left-hand corner of the main menu), and click on the Fetch/Validate link next to the ORCID field.  This will take you to the ORCID site and allow you to create a new iD or authenticate a pre-existing iD in Editorial Manager

(7) Update your Profile Information: Now that your manuscript has been provisionally accepted, please log into Editorial Manager and update your profile, if needed. Go to https://www.editorialmanager.com/ppathogens, log in, and click on the "Update My Information" link at the top of the page. Please update your user information to ensure an efficient production and billing process. 

(8) LaTeX users only: Our staff will ask you to upload a TEX file in addition to the PDF before the paper can be sent to typesetting, so please carefully review our Latex Guidelines http://journals.plos.org/plospathogens/s/latex in the meantime.

(9) If you have associated protocols in protocols.io, please ensure that you make them public before publication to guarantee immediate access to the methodological details.

Best regards,

Yasuko Mori

Guest Editor

PLOS Pathogens

Erik Flemington

Section Editor

PLOS Pathogens

Kasturi Haldar

Editor-in-Chief

PLOS Pathogens

orcid.org/0000-0001-5065-158X

Grant McFadden

Editor-in-Chief

PLOS Pathogens

orcid.org/0000-0002-2556-3526
---

## [Editor Report · Acceptance letter]

21 Nov 2019

Dear Dr Pasdeloup,

We are delighted to inform you that your manuscript, "Dynamic organization of Herpesvirus glycoproteins on the viral envelope revealed by super-resolution microscopy," has been formally accepted for publication in PLOS Pathogens.

Best regards,

Kasturi Haldar

Editor-in-Chief

PLOS Pathogens

orcid.org/0000-0001-5065-158X

Grant McFadden

Editor-in-Chief

PLOS Pathogens

orcid.org/0000-0002-2556-3526